# Effect of Statins on All-Cause Mortality in Adults: A Systematic Review and Meta-Analysis of Propensity Score-Matched Studies

**DOI:** 10.3390/jcm11195643

**Published:** 2022-09-25

**Authors:** Marcin M. Nowak, Mariusz Niemczyk, Michał Florczyk, Marcin Kurzyna, Leszek Pączek

**Affiliations:** 1Department of Pulmonary Circulation, Thromboembolic Diseases and Cardiology at the European Health Center, 05-400 Otwock, Poland; 2Department of Immunology, Transplant Medicine and Internal Diseases, Medical University of Warsaw, 02-006 Warsaw, Poland

**Keywords:** statins, all-cause mortality, cardiovascular disease, non-cardiovascular disease

## Abstract

Statins are lipid-lowering medications used for the prevention of cardiovascular disease (CVD), but the pleiotropic effects of statins might be beneficial in other chronic diseases. This meta-analysis investigated the association between statin use and mortality in different chronic conditions. Eligible studies were real-world studies that compared all-cause mortality over at least 12 months between propensity score-matched statin users and non-users. Overall, 54 studies were included: 21 in CVD, 6 in chronic kidney disease, 6 in chronic inflammatory diseases, 3 in cancer, and 18 in other diseases. The risk of all-cause mortality was significantly reduced in statin users (hazard ratio: 0.72, 95% confidence interval: 0.66–0.76). The reduction in mortality risk was similar in CVD studies (0.73, 0.66–0.76) and non-CVD studies (0.70, 0.67–0.79). There were no significant differences in the risk reduction between cohorts with different diseases (*p* = 0.179). The greatest mortality reduction was seen in studies from Asia (0.61, 0.61–0.73) and the lowest in studies from North America (0.78, 0.73–0.83) and Australia (0.78, 0.62–0.97). There was a significant heterogeneity (I2 = 95%, tau2 = 0.029, *p* < 0.01). In conclusion, statin use was associated with a significantly reduced risk of all-cause mortality in real-world cohorts with CVD and non-CVD.

## 1. Introduction

Statins are lipid-lowering medications used for the primary and secondary prevention of cardiovascular disease (CVD) [1,2]. Apart from lowering blood lipid levels, statins might reduce cardiovascular risk through antithrombotic and anti-inflammatory effects, atherosclerotic plaque regression, or improved endothelial function [3]. Some of these pleiotropic effects might be beneficial also in patients without CVD. The benefits of statin use were reported for patients with nephropathy, head injury, rheumatoid arthritis, neurodegenerative diseases, cancer, or infections [4,5].

A reduction in all-cause mortality is a reliable measure of treatment efficacy in various populations. Therefore, all-cause mortality seems to be an adequate outcome to compare the benefits of statins between patients with different diseases. Some meta-analyses of randomized controlled trials (RCTs) found that statins reduced all-cause mortality when used in the primary and secondary prevention of CVD [6,7], heart failure [8], atrial fibrillation [9], and chronic kidney disease (CKD) requiring dialysis [10]. In contrast, other meta-analyses did not find any significant effect of statins on all-cause mortality in CKD [11] or the primary prevention of CVD [12,13]. To date, no RCTs have evaluated the effects of statins on mortality in other chronic diseases.

Considering the inconsistent evidence from RCTs on the association between statin use and all-cause mortality in patients with CVD, and the lack of evidence for numerous other diseases, we reviewed nonrandomized studies that used propensity score matching to reduce confounding. Such quasi-experimental studies provide real-world evidence, which helps assess the efficacy of treatments in nonhomogeneous populations that are characteristic of clinical practice. Moreover, these studies can provide relevant information when RCTs are unavailable [14]. The purpose of this study was to analyze the association between statin use and all-cause mortality in different chronic diseases in studies that used propensity score matching to match statin users and non-users in a real-world setting.

## 2. Methods

### 2.1. Protocol and Registration

The systematic review protocol was developed in accordance with the Preferred Reporting Items for Systematic Review and Meta-analysis Protocols (PRISMA) guidance [15]. The protocol is available in supplement 1.

### 2.2. Data Sources and Searches

This was a systematic review and meta-analysis. The PubMed electronic database was searched for articles published in English from February 2012 to February 2022. Last search was performed 20th February 2022. Relevant keywords were applied alone or in combination to identify data. The search strategy with the number of hits is presented in the study protocol. For abstracts potentially meeting the inclusion criteria, full-text publications were retrieved. Each study was assessed for eligibility by two independent reviewers, according to the criteria presented in the study protocol. Reasons for exclusion were briefly documented.

### 2.3. Study Selection

Studies were eligible that reported adjusted hazard ratios (HR) for all-cause mortality over at least 12 months in statin users vs. non-users in real-world cohorts matched with propensity score matching. Only studies carried out among adults were included.

### 2.4. Data Extraction and Quality Assessment

Two independent investigators (MMN and MN) extracted the following variables: adjusted HR, sample size, percentage of men, mean age, average follow-up (mean or median), and number of deaths. Disagreements were resolved by consensus. The total duration of follow-up was obtained from publications or calculated by multiplying the average follow-up by cohort size (patient-years). The number of deaths per 1000 patient-years was obtained from publications or calculated by dividing the number of deaths by the total follow-up duration. 

Risk of bias assessment of the included studies was performed by two independent authors using the Newcastle-Ottawa Scale (NOS). NOS consists of three domains: (1) selection, (2) comparability, and (3) outcome [16]. Discrepancies were resolved by discussion. The certainty of evidence was assessed based on Grading of Recommendations, Assessment, Development and Evaluations (GRADE) framework.

### 2.5. Data Synthesis and Analysis

We conducted a meta-analysis of adjusted HRs for changes in all-cause mortality in statin users. Standard errors were calculated from 95% confidence intervals or from *P* values [17]. The log-transformed values of point estimates and of standard errors were used in an inverse variance random-effects meta-analysis, with the restricted maximum-likelihood estimator for tau^2^ and the Q-profile method for the confidence interval of tau^2^ and tau. Heterogeneity was expressed with the *I*^2^ statistic and evaluated with the Cochran’s Q test. Prediction intervals were calculated to aid the interpretation of the estimates, with consideration of heterogeneity [18]. Influential studies with the greatest impact on the estimate and heterogeneity were explored by a visual inspection of the Baujat plot [19]. Subgroup analyses were performed to compare the cohorts of patients with CVD vs. those with other diseases. CVD was defined as coronary artery disease, ischemic stroke, peripheral artery disease, heart failure, atrial fibrillation, and valvular disease. Additional subgroup analyses compared cohorts with CVD, CKD, inflammatory diseases (autoimmune diseases, gout), cancer, and other diseases. Studies including patients with both CVD and CKD were classified as CVD cohorts. We also compared studies conducted on different continents. A restricted maximum-likelihood random-effects meta-regression analysis was used to explore heterogeneity, with the following covariates assessed: percentage of men, mean age, publication year, average follow-up, and number of deaths per 1000 person-years. A funnel plot and the Egger’s test were used to assess publication bias [20]. A *P* value of less than 0.05 was considered statistically significant. The R software (version 4.1.2) and the *meta* and *dmetar* packages were used for all analyses [21,22]. The study did not receive any funding.

## 3. Results

A total of 665 citations were identified, and 210 potentially eligible articles were retrieved in full text. Overall, 54 studies were included in the review (Figure 1) [23,24,25,26,27,28,29,30,31,32,33,34,35,36,37,38,39,40,41,42,43,44,45,46,47,48,49,50,51,52,53,54,55,56,57,58,59,60,61,62,63,64,65,66,67,68,69,70,71,72,73,74,75,76]. Risk of bias assessment using NOS showed that the studies have low risk of bias. The characteristics of included studies are summarized in Table 1. In the analysis, we included 3 cohorts from the study by Lee et al. [74] The mean age of patients was 67.3 years (range, 33.4–85.2), and men constituted 58% of the population. A total of 21 studies (23 cohorts) were focused on CVD. Among the remaining 33 non-CVD studies, 6 included patients with CKD; 6 with inflammatory diseases; 3 with cancer; and 18 with other diseases. Almost half of the studies (*n* = 25) were conducted in Asia; 17 in North America; 12 in Europe; and 2 in Australia.

### 3.1. Statin Use and All-Cause Mortality

The pooled estimate showed that statin use was associated with a significant reduction in all-cause mortality (HR = 0.72; 95% CI, 0.68–0.76), but there was significant heterogeneity (*I^2^* = 95%, tau^2^ = 0.0294, *p* < 0.01; prediction interval, 0.51–1.02; Figure 2).

### 3.2. Publication Bias

The funnel plot was asymmetrical, with more studies reporting estimates that were lower than the pooled estimate (Figure 3). The Egger’s test showed a trend for a publication bias (*p* = 0.096). 

### 3.3. Influential Studies

The Baujat plot (Figure 4) revealed six studies with a substantial contribution to the overall heterogeneity and influence on the pooled result. After excluding these studies, the pooled effect estimate was 0.73 (95% CI, 0.70–0.76), and the residual heterogeneity was lower but significant (*I*^2^ = 72%, tau^2^ = 0.0123, *p* < 0.01; prediction interval, 0.58–0.92). 

### 3.4. Sensitivity Analyses

There was no significant difference in statin-related reduction in the risk of mortality between CVD and non-CVD cohorts (Table 2). The estimate was lower in cohorts with inflammatory diseases and cancer than in the remaining subgroups, but the difference was not significant (Table 2). The estimates were lower in Asian and European studies than in those conducted in Australia and North America (Table 2, *p* = 0.044). 

### 3.5. Meta-Regression Analysis

After the exclusion of influential studies, the HRs for the association between statin use and mortality tended to increase with mean age (*p* = 0.075) and to decrease with average follow-up (*p* = 0.051; Table 3, Figure 5). The HRs were not associated with the percentage of men, publication year, and number of deaths per 1000 patient-years (Table 3, Figure 5). None of the covariates accounted for heterogeneity (Table 3).

## 4. Discussion

This systematic review evaluated evidence from real-world studies assessing statin use, with a total of over 4 million patient-years of follow-up. All studies except one reported lower all-cause mortality in statin users vs. matched cohorts. The meta-analysis showed that statin use was associated with a significant reduction in all-cause mortality, with the risk lower by about 30% in patients on statins. There was no substantial difference in mortality reduction between studies including CVD and non-CVD cohorts. Substantial heterogeneity was noted in the main analysis, subgroup analyses, and after the exclusion of influential studies. None of the covariates assessed in meta-regression models accounted for heterogeneity. The funnel plots suggested possible publication bias towards studies reporting favorable effects of statins, which may be study limitation, although this effect was not significant. Therefore, although the included studies used propensity score matching and adjusted the estimates for potential confounders, the overall evidence on the association of statins with all-cause mortality in real-world clinical practice is of suboptimal quality. 

Statins are potent antiatherogenic agents that reduce blood lipid concentrations and prevent atherosclerosis progression [77]. These effects likely translate into a lower risk of cardiovascular events, including death. There is evidence that statins might also have nephroprotective effects and slow the progression of CKD (a slower decline in glomerular filtration rate among statin users) [78]. However, other studies found that high-dose statin therapy, as compared with low-dose therapy, was associated with an increased risk of kidney disease [79]. The anti-inflammatory and antioxidant effects of statins can be beneficial in patients with autoimmune diseases [80]. Statins might also slow the progression of certain types of cancer by inhibiting cancer cell proliferation and tumor angiogenesis and by inducing apoptosis and stimulating immune surveillance of cancer cells [81]. 

In our meta-analysis, the reduction in all-cause mortality in statin users was similar among those with or without CVD. Assuming there was a lower cardiovascular risk in non-CVD than CVD cohorts, a similar risk reduction in mortality in the two cohort types, which we found in this meta-analysis, may suggest that statins reduce the risk of mortality through non-cardiovascular effects as well. Among patients with CKD, the reduction in all-cause mortality in statin users was similar to that observed in CVD patients. CVD is common in patients with CKD and constitutes the most frequent cause of death in this population [82]. Thus, statin-associated reduction in morality risk in CKD is likely due to a lower incidence of cardiovascular events. Further subgroup analyses suggested that statin-associated reduction in mortality risk could be lower in inflammatory diseases and cancer. In previous studies, statin use was associated with lower disease severity scores in rheumatoid arthritis [83]. However, it remains unknown whether this disease-modifying effect translates into lower mortality, because CVD is a common cause of death in patients with rheumatoid arthritis [84]. In our study, the lower survival benefit in statin users among patients with cancer or inflammatory disease, compared with other subgroups, could be due to a lower cardiovascular risk in these patients. Similar to our study, a meta-analysis of studies enrolling over 1 million patients with cancer found that statin use was associated with a significant decrease in all-cause mortality (HR = 0.70) [85]. Notably, statin use was also associated with reduced cancer-specific mortality (HR = 0.60) [85].

We observed that the reduction in all-cause mortality was lower in the United States and Australia than in Europe and Asia. Pharmaceutical sales data show that statins are used more often in the United States and Australia than in Europe and Asia [86]. This is likely due to a higher number of prescriptions in primary prevention, that is, among patients with lower cardiovascular risk in whom the survival benefit is lower than in patients requiring secondary prevention [87]. This could explain a lower mortality reduction in American and Australian studies. 

The existing evidence from observational studies shows that statins are associated with a significantly reduced all-cause mortality, whereas this effect is not always significant in RCTs. Limited sample size and shorter follow-up in RCTs may partly explain this observation. Indeed, we found that the reduction in all-cause mortality in statin users tended to increase with longer follow-up. Moreover, among observational studies, we found a trend for a publication bias towards studies reporting favorable outcomes.

Our study is limited by substantial heterogeneity, with the upper limit of the prediction interval above 1. However, the sensitivity analysis after exclusion of influential studies was characterized by reduced heterogeneity, with the upper limit of the prediction interval below 1. Heterogeneity in meta-analyses of large registry-based studies is typically greater than that in meta-analyses of RCTs [88,89,90,91]. It is possible that in our meta-analysis substantial heterogeneity occurred because individual studies included participants who differed in terms of statin type, statin dose, adherence, treatment duration, and comorbidities. In RCTs, such confounding factors are balanced by random allocation to treatment, but they are difficult to be accounted for in observational studies, even when propensity score matching is used. From a statistical standpoint, large studies have a low sampling error (high precision), and therefore, the differences in effect size between such studies result in high heterogeneity. Although the covariates assessed in meta-regression models did not account for heterogeneity, we observed that the reduction in all-cause mortality in statin users tended to be greater in studies with longer follow-up and to be lower with the increasing age of participants. This observation is consistent with previous reports [90,92] and underscores the importance of conducting real-world studies that reflect clinical practice, have a much longer follow-up, and have a greater proportion of elderly patients than that in most clinical trials. The strengths of our study include sensitivity analyses performed after the exclusion of influential studies and in different subgroups. Moreover, we carried out meta-regression analyses and used all-cause mortality as an outcome that is unlikely to be biased. Searching for eligible studies in the PubMed database only is another limitation.

In conclusion, statin use was associated with a significant reduction in all-cause mortality in various populations treated in real-world clinical practice in an analysis of over 4 million patient-years. It remains unclear whether the reduction in mortality risk associated with statins is solely due to a reduced incidence of cardiovascular death or other effects of these medications. Substantial heterogeneity limits the available evidence on the association between statin use and all-cause mortality in real-world practice.

## Figures and Tables

**Figure 1 jcm-11-05643-f001:**
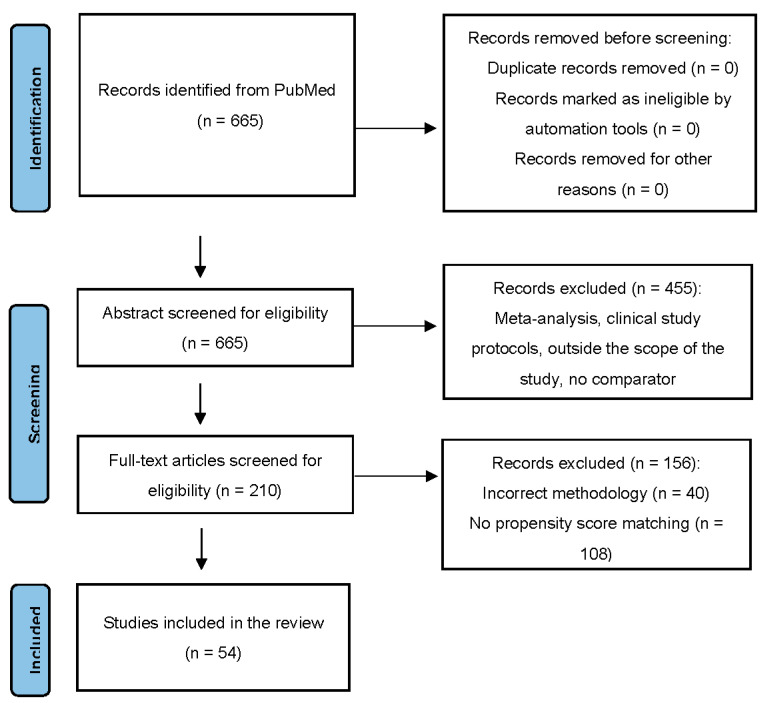
Preferred Reporting Items for Systematic Reviews and Meta-Analyses (PRISMA) flow chart of the study selection process.

**Figure 2 jcm-11-05643-f002:**
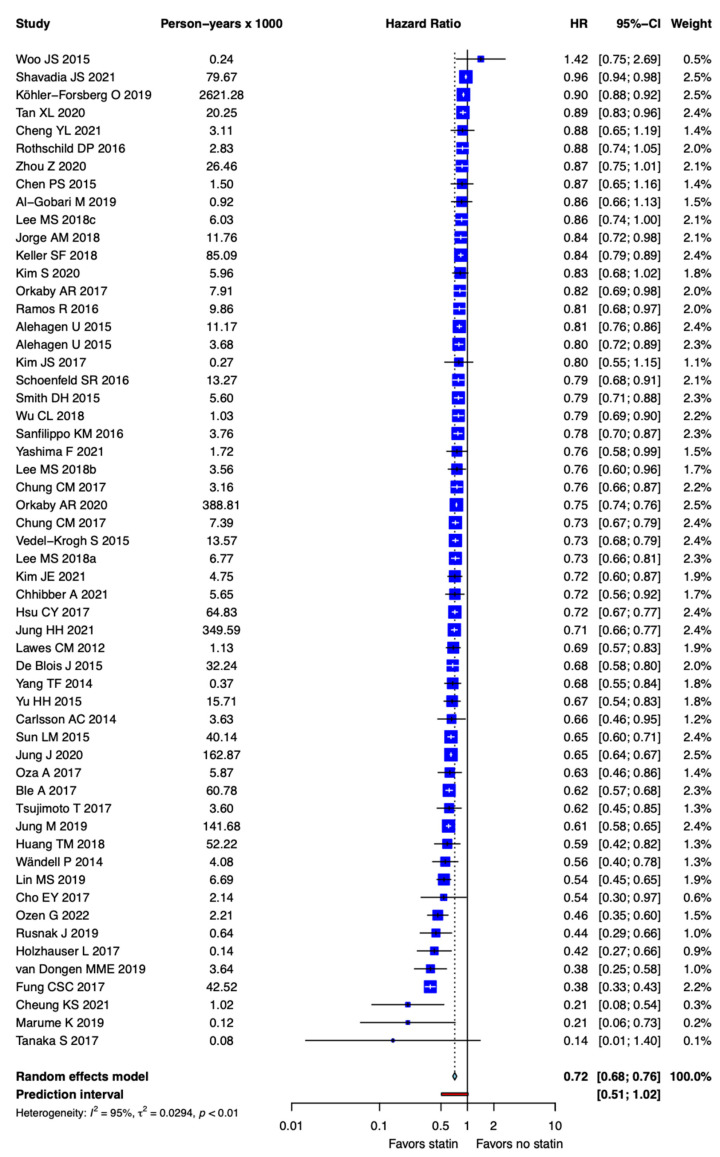
Forest plot showing the association between statin use and all-cause mortality.

**Figure 3 jcm-11-05643-f003:**
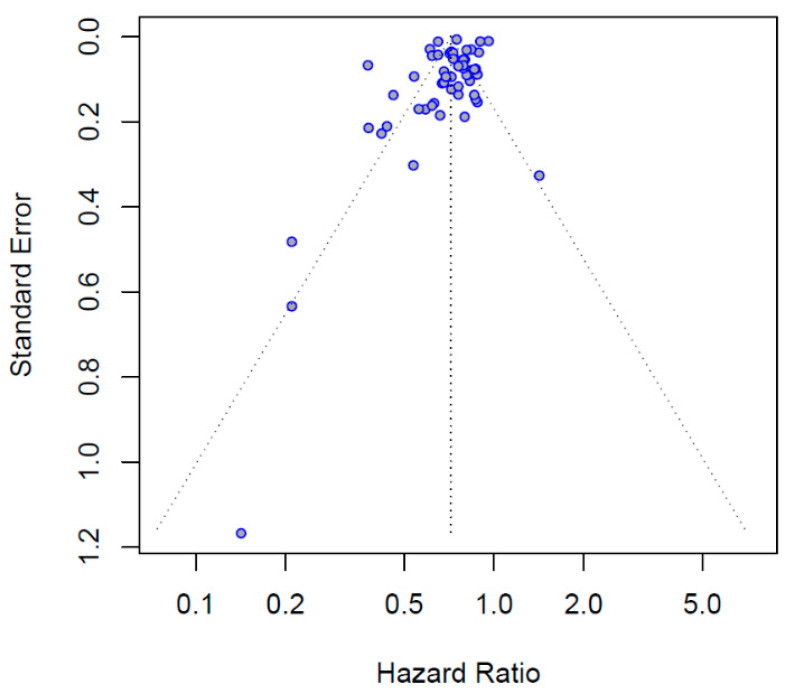
Funnel plot for publication bias.

**Figure 4 jcm-11-05643-f004:**
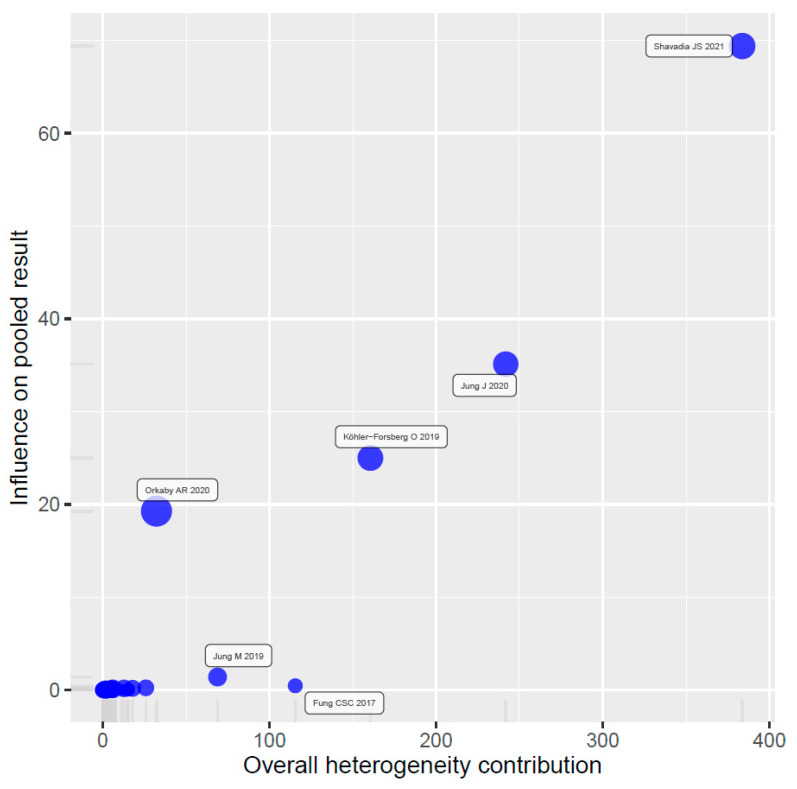
Baujat plot for the analysis of influential studies.

**Figure 5 jcm-11-05643-f005:**
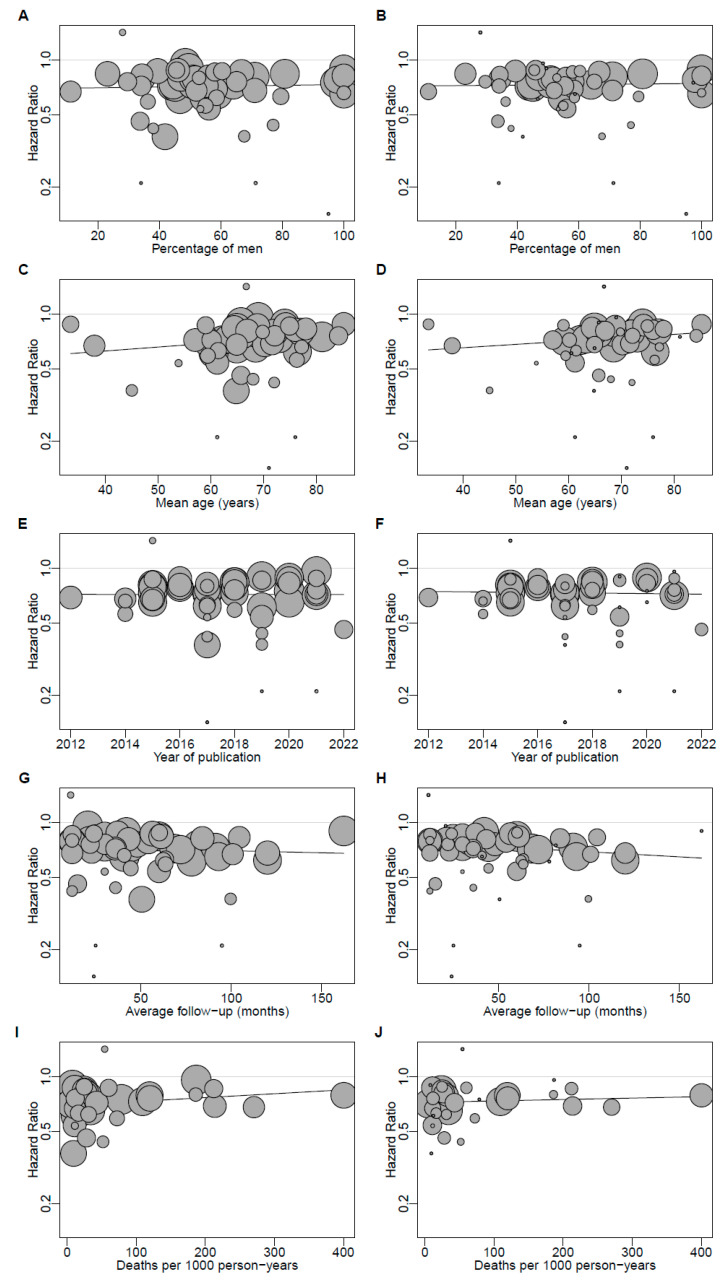
Bubble plots for the association between statin use and all-cause mortality regressed against the percentage of men (**A**,**B**), mean age (**C**,**D**), year of publication (**E**,**F**), average follow-up (**G**,**H**), and number of deaths per 1000 patient-years (**I**,**J**). Plots on the left represent the primary meta-analysis, whereas plots on the right represent the meta-analysis after the exclusion of influential studies. Bubble size represents study weight.

**Table 1 jcm-11-05643-t001:** Study characteristics.

Author and Year	Continent	Population	Mean Age (Years)	Sex (% Male)	Adjusted HR for All-Cause Mortality	Mean/Median Follow-up (Months)	NOS
Ozen 2022 [23]	North America	Inflammatory disease	65.7	33.7	0.46	15.0	7
Kim 2021 [24]	Asia	CKD	57.0	55.4	0.72	35.7	8
Yashima 2021 [25]	Asia	CVD	84.2	29.6	0.76	22.0	7
Cheng 2021 [26]	Asia	CKD	33.4	45.6	0.88	60.2	8
Cheung 2021 [27]	Asia	Other	61.2	71.3	0.21	94.8	8
Chhibber 2021 [28]	North America	Inflammatory disease	60.1	34.1	0.72	36.0	8
Jung 2021 [29]	Asia	Other	59.2	45.1	0.71	72.0	8
Shavadia 2021 [30]	North America	CVD	69.0	48.4	0.96	20.7	7
Kim 2020 [31]	Asia	Other	78.0	34.3	0.83	104.4	8
Tan 2020 [32]	North America	Cancer	74.0	100	0.89	42.0	8
Lin 2018 [33]	Asia	CVD	61.2	56.1	0.54	60.0	8
Köhler-Forsberg 2019 [34]	Europe	Other	65.7	49.5	0.90	162.2	8
Jung 2019 [35]	Asia	Other	60.5	46.7	0.61	78.0	8
Huang 2018 [36]	Asia	CKD	59.4	36.2	0.59	63.6	8
Jorge 2018 [37]	North America	Inflammatory disease	64.4	23.1	0.84	61.2	8
Keller 2017 [38]	North America	Inflammatory disease	64.9	80.8	0.84	60.0	8
Orkaby 2017 [39]	North America	Other	76.0	100	0.82	84.0	8
Kim 2017 [40]	Asia	CVD	69.8	52.9	0.80	12.0	7
Oza 2017 [41]	North America	Inflammatory disease	61.4	79.5	0.63	63.6	8
Fung 2017 [42]	Asia	Other	64.8	41.8	0.38	50.5	8
Chung 2017 [43]	Asia	CKD	63.3	44.5	0.73	44.4	8
Hsu 2017 [44]	Asia	CVD	62.2	44.0	0.72	68.4	8
Holzhauser 2017 [45]	North America	Other	72.0	38.0	0.42	12.0	7
Cho 2017 [46]	Asia	CKD	53.9	53.3	0.54	30.0	8
Sanfilippo 2016 [47]	North America	Cancer	68.6	97.9	0.78	34.0	8
Tanaka 2017 [48]	Asia	CVD	71.0	95.0	0.14	24.0	8
Rothschild 2016 [49]	North America	CVD	85.2	46.0	0.88	37.2	8
Ble 2017 [50]	Europe	CVD	76.4	54.5	0.62	120.0	8
Ramos 2016 [51]	Europe	Other	66.9	55.9	0.81	43.2	8
Woo 2015 [52]	Asia	CVD	66.7	28.0	1.42	11.2	7
Vedel-Krogh 2015 [53]	Europe	Other	71.0	64.0	0.73	91.2	8
Sun 2015 [54]	Asia	Cancer	68.5	100	0.65	93.0	8
Yu 2015 [55]	Asia	Inflammatory disease	37.9	11.1	0.67	100.8	8
Schoenfeld 2016 [56]	North America	Inflammatory disease	65.3	34.4	0.79	54.1	8
Alehagen 2015 [57]	Europe	CVD	77.0	47.0	0.80	21.3	7
Smith 2015 [58]	North America	Other	72.0	50.9	0.79	12.0	7
Chen 2015 [59]	Asia	CVD	59.0	60.4	0.87	24.0	8
Alehagen 2015 [60]	Europe	CVD	73.0	71.0	0.81	24.9	7
De Blois 2015 [61]	Europe	CVD	70.2	71.0	0.68	120.0	8
Yang 2014 [62]	Asia	Other	64.9	52.0	0.68	12.0	7
Carlsson 2014 [63]	Europe	CVD	77.2	100	0.66	40.8	8
Wändell 2014 [64]	Europe	CVD	76.3	55.0	0.56	44.4	8
Lawes 2012 [65]	Australia	Other	71.6	58.4	0.69	22.8	7
Orkaby 2020 [66]	North America	Other	81.1	97.3	0.75	81.6	8
Zhou 2020 [67]	Australia	Other	74.2	39.4	0.87	56.4	8
Jung 2020 [68]	Asia	CKD	64.9	58.8	0.65	40.8	8
van Dongen 2019 [69]	Europe	CVD	45.0	67.6	0.38	99.6	8
Rusnak 2019 [70]	Europe	CVD	68.0	77.0	0.44	36.0	8
Al-Gobari 2019 [71]	Europe	CVD	74.9	58.0	0.86	12.0	7
Marume 2019 [72]	Asia	CVD	76.0	34.0	0.21	25.0	8
Wu 2018 [73]	Asia	Other	66.3	51.6	0.79	12.0	7
Lee 2018a [74]	North America	CVD	73.0	50.6	0.73	30.0	8
Lee 2018b [74]	North America	CVD	72.1	65.1	0.76	30.0	8
Lee 2018c [74]	North America	CVD	68.8	66.7	0.86	30.0	8
Chung 2017 [75]	Asia	CVD	66.0	58.0	0.76	48.0	8
Tsujimoto 2017 [76]	Asia	Other	nd	58.6	0.62	62.4	8

CKD, chronic kidney disease; CVD, cardiovascular disease; DM, diabetes mellitus; HR, hazard ratio; nd, no data; NOS, Newcastle-Ottawa Scale. Inflammatory diseases consist of rheumatoid arthritis, systemic lupus, gout, and ankylosing spondylitis. Other include the elderly, hepatitis, lung diseases, diabetes mellitus, and healthy subjects.

**Table 2 jcm-11-05643-t002:** Sensitivity analyses of the association between statin use and all-cause mortality.

	All Studies (*n* = 56)						Without Influential Studies (*n* = 50)					
Subgroup	No. of Studies	Estimate (HR)	95% CI	Prediction Interval	I^2^ (%)	*p* Value	No. of Studies	Estimate (HR)	95% CI	Prediction Interval	*I*^2^ (%)	*p* Value
**CVD**	23	0.73	0.66–0.76	0.51–1.04	92	0.598	22	0.72	0.66–0.78	0.53–0.97	73	0.497
**Non-CVD**	33	0.70	0.67–0.79	0.49–1.02	95		28	0.74	0.71–0.78	0.60–0.91	71	
**CVD**	23	0.73	0.67–0.79	0.51–1.04	92	0.179	22	0.72	0.66–0.77	0.53–0.97	73	0.525
**CKD**	6	0.69	0.63–0.75	0.55–0.85	61		5	0.72	0.67–0.77	0.64–0.81	1.2	
**Inflammatory disease**	6	0.78	0.72–0.85	0.63–0.96	39		6	0.78	0.72–0.85	0.63–0.96	39	
**Cancer**	3	0.77	0.64–0.92	0.07–7.52	94		3	0.77	0.64–0.92	0.07–7.52	94	
**Other**	18	0.68	0.60–0.76	0.40–1.13	96		14	0.72	0.66–0.78	0.54–0.95	67	
**Asia**	25	0.67	0.61–0.73	0.45–0.98	83	0.044	22	0.71	0.67–0.74	0.62–0.80	51	0.126
**Australia**	2	0.78	0.62–0.97	-	73		2	0.78	0.62–0.97	-	73	
**Europe**	12	0.71	0.62–0.79	0.46–1.06	92		11	0.69	0.61–0.77	0.47–1.00	79	
**North America**	17	0.78	0.73– 0.83	0.60–1.02	96		15	0.78	0.72–0.83	0.61–0.98	68	

CI, confidence interval; CKD, chronic kidney disease; CVD, cardiovascular disease; HR, hazard ratio. *p* values were derived from the Q test for subgroup differences.

**Table 3 jcm-11-05643-t003:** Meta-regression models for the association between statin use and all-cause mortality.

	Primary Meta-Analysis					Omitting Influential Studies				
	No. of Studies	Estimate	95% CI	*p* Value	Residual *I*^2^ (%)	No. of Studies	Estimate	95% CI	*p* Value	Residual *I*^2^ (%)
**Percentage of men**	56	0.0005	−0.0021–0.0031	0.698	95	50	0.0003	−0.0017–0.0024	0.738	72
**Mean age (years)**	56	0.0050	−0.0009–0.0109	0.094	95	55	0.0042	−0.0004–0.0089	0.075	70
**Publication year**	56	−0.0003	−0.0240–0.0235	0.983	96	50	−0.0034	−0.0228–0.0160	0.729	73
**Average follow-up (months)**	56	−0.0005	−0.0021–0.0011	0.520	96	50	−0.0013	−0.0026–0.0000	0.051	67
**Deaths per 1000 person-years**	35	0.0005	−0.0003–0.0013	0.204	96	30	0.0002	−0.0004–0.0007	0.514	68

CI, confidence interval.

## Data Availability

The data that support the findings of this study are available on request from the corresponding author.

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
