# Peer review of "Effect of Statins on All-Cause Mortality in Adults: A Systematic Review and Meta-Analysis of Propensity Score-Matched Studies"

_jcm, 2022, doi:10.3390/jcm11195643_

Round 1

Reviewer 1 Report

The abstract should be more precisely concluded.

The results must be interpreted in more detail referring o the particular diseases and/or factors analyzed. This section sounds just a statistical description.

Discussion is missing. Conclusions perhaps replace Discussion and are inconclusive and vague. 

Author Response

The abstract should be more precisely concluded.

We have re-written the abstract altogether for brevity and clarity:

“Statins are lipid-lowering medications used for the prevention of cardiovascular disease (CVD), but the pleiotropic effects of statins might be beneficial in other chronic diseases. This meta-analysis investigated the association between statin use and mortality in different chronic conditions.  Eligible were real-world studies that compared all-cause mortality over at least 12 months between propensity score-matched statin users and non-users. Overall, 54 studies were included: 21 in CVD, 6 in chronic kidney disease, 6 in chronic inflammatory diseases, 3 in cancer, and 18 in other diseases. The risk of all-cause mortality was significantly reduced in statin users (hazard ratio: 0.72, 95% confidence interval: 0.66-0.76). The reduction in mortality was similar in CVD studies (0.73, 0.66-0.76) and non-CVD studies (0.70, 0.67-0.79). There were no significant differences in the risk reduction between cohorts with different diseases (p = 0.179). The greatest mortality reduction was seen in studies from Asia (0.61, 0.61- 0.73) and the lowest in studies from North America (0.78, 0.73- 0.83) and Australia (0.78, 0.62- 0.97). There was a significant heterogeneity (I2 = 95%, tau2 = 0.029, P < 0.01). In conclusion, statin use was associated with a significantly reduced risk of all-cause mortality in real-world cohorts with CVD and non-CVD.“

The results must be interpreted in more detail referring to the particular diseases and/or factors analyzed. This section sounds just a statistical description.

Per the PRISMA reporting guidelines for systematic reviews and meta-analyses, we present the statistical analyses in the results section, including pooled estimates, sensitivity analyses, and heterogeneity analyses. We classified the studies included in the metanalysis in 6 categories: cardiovascular diseases, chronic kidney disease, inflammatory diseases, cancer, and other. Further subclassification seems unfeasible due to a limited number of studies. For reference, we describe each individual study in Table 1 and give the estimates for all studies in Fig. 2. 

Our results are then interpreted in the discussion section, as per the PRISMA guidelines. We discuss our analyses with previous studies carried out in different cohorts with cardiovascular disease, chronic kidney disease, chronic inflammatory disease, and cancer. We made an effort to include all relevant studies from our search (we cite 92 studies in total). Further, we speculate on the mechanisms that could confer favorable effects of statins in different diseases.

Discussion is missing. Conclusions perhaps replace Discussion and are inconclusive and vague. 

Indeed. We used the wrong heading: “conclusions” instead of “discussion”. This was corrected accordingly. We tried to present the discussion in the standard way: summing up our results in the first paragraph and then referring our results to previous research, with limitations and conclusions presented at the end. As suggested, we elaborated on the conclusions:

“In conclusion, statin use was associated with a significant reduction in all-cause mortality in various populations treated in real-world clinical practice in an analysis of over 4 million patients-years. It remains unclear whether the reduction in mortality associated with statins is solely due to a reduced incidence of cardiovascular death or other effects of these medications. Substantial heterogeneity limits the available evidence on the association between statin use and all-cause mortality in real-world practice.“

Reviewer 2 Report

The hypothesis and the purpose of the study are not clear. The inclusion criteria are not clear. The conclusion is not well expressed. Difficult manuscript to be followed.

Author Response

The hypothesis and the purpose of the study are not clear. The inclusion criteria are not clear. The conclusion is not well expressed. Difficult manuscript to be followed.

As a meta-analyses, this study did not have an a priori hypothesis but aimed to summarize the existing evidence. In the introduction, we formulated the purpose of the study more clearly.

“The purpose of this study was to analyze the association between statin use and all-cause mortality in different chronic diseases in studies that used propensity score matching to match statin users and non-users in a real-world setting”.

We specified the inclusion criteria more clearly in the methods: “Eligible were studies that reported adjusted hazard ratios (HR) for all-cause mortality over at least 12 months in statin users vs. non-users in real-world cohorts matched with propensity score matching. Included were only studies carried out among adults”.

We agree that the results reported in the manuscript are complex. We included over 50 studies in this meta-analysis, whereas a typical meta-analysis includes ~6 studies (see Davey et al. BMC Med Res Methodol. 2011 Nov 24;11:160). In addition, because of substantial heterogeneity, we carried out additional analyses, such as sensitivity analyses in many subgroups, role of influential studies, meta-regressions, to find potential sources of heterogeneity. For an easier comprehension, we presented the results in structured tables and plots. We believe these additional analyses add to the value of our study.

Reviewer 3 Report

The work is undoubtedly interesting and the analyzed problem is still very relevant.

I recommend changing the paper structure. Much of the text you label "Conclusions" is actually more appropriate for a "Discussion" section, which would be very valuable for this work.

The work will certainly arouse the interest of colleagues and the publication could be useful in further research of the topic.

Author Response

The work is undoubtedly interesting and the analyzed problem is still very relevant.

I recommend changing the paper structure. Much of the text you label "Conclusions" is actually more appropriate for a "Discussion" section, which would be very valuable for this work.

The work will certainly arouse the interest of colleagues and the publication could be useful in further research of the topic.

The heading was changed from “conclusions” to “discussion” as suggested.

Reviewer 4 Report

This is a very interestening and well written paper, and it was a pleasure to review it.

The paper reports a meta-analysis of observational (propensity score-matched) studies specifically addressing the effect of statins on all-cause mortality. Study strengths are inclusion of only studies with (propensity score) adjusted data and real life populations; limitations are inherent to observational studies and derivative high heterogeneity.

The idea (real life populations; all-cause mortality endopoint not captured in RCTs) is intriguing and certainly mertis publishing, there are, however, some minor issues that may be addressed.

1) The heterogeneity is indeed very high and expectedly so — the observational studies included diverse populations, statins, statin doses, therapy durations etc. I am not entirely sure that simply excluding the influential studies is OK — rather, the exclusion may be considered as a exploratory/sensitivity analysis. Please, tone down the discussion in this respect just a little.

2) Can the authors comment (in the discussion) why they believe a meta-analysis of observational studies suggest an effect of all-cause mortality, whereas most RCT do not? e.g. all-cause mortality associated with statin therapy is driven by CV mortality reduction (as per our current understanding), than it all comes down to effect size, which might be overblown by observational studies-derived (large) effect size and publication bias (only studies reporting on all cause mortality reduction published drive the resuls …).

3) Search selection (Pubmed, English language only, 2012-2022) represents a limitation, i.e. limiting the selection of observational studies to a single source significantly diminishes sensitiviy (for instance, see Lameshow et al, J Clin Epidemiol 2005;58:867-73) - please add to Limitations.

4) The phrase ‘statin-related reduction in mortality’ should probably be rephrased to ‘statin-related relative reduction mortality risk’ (e.g., Line 27)

5) line 223 and 224; how do you reconcile the comparable risk reduction in CVD and non-CVD patients with the hypothesis that risk reduction is smaller in primary prevention American and Austraila populations?

4) There are some typos — e.g. 217 statin (instead of stain), language use is otherwise excellent

Author Response

This is a very interesting and well written paper, and it was a pleasure to review it. The heterogeneity is indeed very high and expectedly so — the observational studies included diverse populations, statins, statin doses, therapy durations etc. I am not entirely sure that simply excluding the influential studies is OK — rather, the exclusion may be considered as a exploratory/sensitivity analysis. Please, tone down the discussion in this respect just a little.

As suggested, we now refer to the analysis that excluded the influential studies as a sensitivity analysis: “However, the sensitivity analysis after exclusion of influential studies was characterized by reduced heterogeneity, with the upper limit of the prediction interval below 1.”

Can the authors comment (in the discussion) why they believe a meta-analysis of observational studies suggest an effect of all-cause mortality, whereas most RCT do not? e.g. all-cause mortality associated with statin therapy is driven by CV mortality reduction (as per our current understanding), than it all comes down to effect size, which might be overblown by observational studies-derived (large) effect size and publication bias (only studies reporting on all cause mortality reduction published drive the resuls …).

This is an interesting insight. We elaborated on this issue as follows: “The existing evidence from observational studies shows that statins are associated with a significantly reduced all-cause mortality, whereas this effect is not always significant in RCTs. Limited sample size and shorter follow-up in RCTs may partly explain this observation. Indeed, we found that the reduction in all-cause mortality in statin users tended to increase with longer follow-up. Moreover, among observational studies, we found a trend for a publication bias towards studies reporting favorable outcomes.”  

Search selection (Pubmed, English language only, 2012-2022) represents a limitation, i.e. limiting the selection of observational studies to a single source significantly diminishes sensitiviy (for instance, see Lameshow et al, J Clin Epidemiol 2005;58:867-73) - please add to Limitations.

We agree and added the following limitation: “Searching for eligible studies in the PubMed database only is another limitation.”

The phrase ‘statin-related reduction in mortality’ should probably be rephrased to ‘statin-related relative reduction mortality risk’ (e.g., Line 27)

This was changed throughout the manuscript.  

lines 223 and 224; how do you reconcile the comparable risk reduction in CVD and non-CVD patients with the hypothesis that risk reduction is smaller in primary prevention American and Austraila populations?

This is and interesting point, which we did not notice ourselves.  We added the following sentence in the discussion: “Assuming there was a lower cardiovascular risk in non-CVD than CVD cohorts, a similar risk reduction in mortality in the two cohort types, which we found in this meta-analysis, may suggest that statins reduce the risk of mortality through non-cardiovascular effects as well”.

There are some typos — e.g. 217 statin (instead of stain), language use is otherwise excellent

This was changed as suggested.

Round 2

Reviewer 1 Report

Now, after considering the updated manuscript I am satisfied with this version.

Author Response

Thank you.